# Cell Mutagenic Autopolyploidy Enhances Salinity Stress Tolerance in Leguminous Crops

**DOI:** 10.3390/cells12162082

**Published:** 2023-08-17

**Authors:** Phetole Mangena

**Affiliations:** Department of Biodiversity, School of Molecular and Life Sciences, Faculty of Science and Agriculture, University of Limpopo, Private Bag X1106, Sovenga 0727, South Africa; phetole.mangena@ul.ac.za or mangena.phetole@gmail.com

**Keywords:** antioxidant activity, autopolyploidy, genome duplication, legumes, molecular and physiological mechanisms, salinity stress, reactive oxygen species

## Abstract

Salinity stress affects plant growth and development by causing osmotic stress and nutrient imbalances through excess Na^+^, K^+^, and Cl^−^ ion accumulations that induce toxic effects during germination, seedling development, vegetative growth, flowering, and fruit set. However, the effects of salt stress on growth and development processes, especially in polyploidized leguminous plants, remain unexplored and scantly reported compared to their diploid counterparts. This paper discusses the physiological and molecular response of legumes towards salinity stress-based osmotic and ionic imbalances in plant cells. A multigenic response involving various compatible solutes, osmolytes, ROS, polyamines, and antioxidant activity, together with genes encoding proteins involved in the signal transduction, regulation, and response mechanisms to this stress, were identified and discussed. This discussion reaffirms polyploidization as the driving force in plant evolution and adaptation to environmental stress constraints such as drought, feverish temperatures, and, in particular, salt stress. As a result, thorough physiological and molecular elucidation of the role of gene duplication through induced autopolyploidization and possible mechanisms regulating salinity stress tolerance in grain legumes must be further studied.

## 1. Introduction

Leguminous crops of the Fabaceae family such as chickpea (*Cicer arietinum*), cowpea (*Vigna unguiculata*), kidney bean (*Phaseolus vulgaris*), lentils (*Lens culinaris*), pea (*Pisum sativum*) and soybean (*Glycine max* L.) are considered the most valuable grain crops playing a key role in the diet of many people throughout the world. Grain legumes, also referred to as pulses, serve as an excellent source of essential proteins, dietary fiber, carbohydrates, mineral nutrients, and phytochemicals with very minimum-to-no fat content [1,2]. These grain crops have formed the basis for early farming culture systems, together with cereal crops centuries ago [3]. They are harvested as both dry seeds and vegetables used for human nutrition and livestock feeding. Apart from their nutritional aspect, species of the Leguminosae remain ecologically dominant across many vegetations [4], fixing atmospheric nitrogen (N) through an association with symbiotic rhizobial bacteria [5] and reducing the need for inorganic N fertilizers. Consequently, whole genome duplication events during the Cretaceous-Paleogene around 66 million years ago played a significant role in legume survival and diversification. This evolutionary process transformed legume wild taxa into forms that enabled sustainable human gains through domestication and numerous breeding programmes [3].

The domestication and production of legumes through agriculture were crucial in safeguarding food and nutritional security while enhancing the environmental nitrogen cycle in terrestrial ecosystems [6,7]. Although N-fixing microbials in association with nodulating plant species mostly grow under arid conditions as companion or dominant species, the degree to which these plants, particularly polyploidized legumes, exhibited tolerance to abiotic stress factors such as drought and salinity stress still requires further investigations. Plant polyploidy (whole-genome duplication) is a common and natural phenomenon that has been considered a speciation mechanism that leads to genome restructuring through either allopolyploidy or autopolyploidy. The occurrence of allo-polyploidized hybrids containing multiple sets of chromosomes derived from different species coincides with major global climatic and geographic changes that led to a period of species mass extinction [3]. According to Rao et al. [8], the variations in gene composition and expression patterns found in homologous polyploid plants resulted in different beneficial physiological responses enhancing tolerance and optimum growth and yield under abiotic stress conditions. The study indicated that autotetraploids of *Lycium ruthenicum* revealed an increased abscisic acid (ABA) content of 78.4% under both normal and severe drought stress conditions compared with their diploid counterparts.

Nevertheless, this study further indicated that molecular and physiological responses of polyploidized *L. ruthenicum* plants showed enhanced upregulation of genes responsible for the biosynthesis and signal transduction pathways of ABA than their diploid counterparts. This sesquiterpenoid hormone plays an important role in many cellular processes such as seed development, dormancy, germination, growth, and development of plants, as well as response to various stresses, and is subjected to complex regulatory mechanisms that involve the actions of a small family of *9-cisepoxycarotenoid dioxygenase* (*NCED*) genes that indirectly produce xanthoxin, which is the main precursor for ABA [9]. The new challenge, however, is to understand the regulation of these biosynthetic genes and their pathways, which predominantly lead to the stimulation of ABA biosynthesis and signal transduction during salinity stress.

Several studies have investigated the effects of abiotic stresses in various crop species of legumes, especially in diploid lines than polyploidized genotypes. As such, more studies must unravel how ABA induces the expression of osmotic proteins, especially by increasing the stress tolerance of plants from genome compositions to gene translational levels, as alluded to by Rao et al. [8]. Under salinity stress, plants generally experience reduced growth, impaired ion or osmotic homeostasis, and disruptions of metabolic processes, subsequently leading to losses in crop growth and yields. This stress serves as one of the major limiting abiotic factors in agriculture, threatening food and nutritional security. The adverse effects of salinity stress constraint in plant cells, organs, and whole plant level take place during any stage of growth, including germination, seedling development, vegetative growth, and reproduction [10]. Furthermore, salinity stress largely influences legume growth and yield by disturbing photosynthesis, hormonal regulation, nutritional imbalances, lowering cell water potential, causing ion toxicity and osmotic stress effects. As reported by Farooq et al. [11], unraveling the response mechanisms and tolerance, as well as exploring the available stress management strategies to this abiotic constraint in grain legumes, may help in the improvement in growth and productivity of leguminous plants growing under salinity stress conditions.

In this paper, the metabolic and genomic effects of salt stress, including tolerance mechanisms of polyploidized plants, in contrast with their diploid counterpart, are discussed. This review analyzed current literature on the application of induced autopolyploidization to enhance plants’ adaptive traits against salinity through a multigenic response involving various compatible solutes, osmolytes, ROS, polyamines, and oxidative stress-linked antioxidant activity, together with genes encoding proteins involved in the signal transduction, regulation, and response mechanisms to this stress. Altogether, the review examined whether synthetic polyploids of grain legumes exhibited enhanced response mechanisms to prevent toxic accumulation of Na^+^ and Cl^−^ ion levels in the cell cytosol through a more complete genomic and physiological understanding, which remains a precondition for the development of new and improved salt-tolerant leguminous varieties.

## 2. Role of Autopolyploidy on Legume Response to Salinity Stress

Plant ploidy level, which refers to the number of copies of the entire genome in a plant cell, has implications for both physiological and molecular responses to both biotic and abiotic stresses. This process involves additional rounds of one or more genome duplications without cytokinesis to generate allo- or auto-polyploidized mutant plants. The ploidy level can also be carried through into the germ line (gametes) to form multiple generations of polyploids [12]. Naturally, autopolyploidy or allopolyploidy can take place due to incomplete meiosis through gametogenesis and the formation of unreduced chromosome numbers by microtubule inhibition within diploid cells [13]. Both meiotic and mitotic cell errors occur spontaneously in plants through variable frequencies depending on the species. This is a widespread phenomenon, occurring naturally and very gradually in flowering plants, but can also be rapidly synthetically induced using mutagenic chemicals such as colchicine, oryzalin, epoxomicin, and nitroxide as described in Figure 1 [14,15,16].

These antimitotic agents are widely reported and comparatively evaluated in literature for their efficiency in inducing chromosome doubling through in vitro and in vivo regeneration of varying plant species [17,18,19,20,21,22,23]. As indicated in Figure 1, the influence of these chemicals on the formation of more than one haploid genome in a single cell nucleus of plants occurs as a result of the interference with kinetochore-microtubule attachments or what is widely known as microtubule depolymerization which is also extensively reported [18,19,23,24]. However, during the production of synthetic polyploids, the efficiency of genome replication depends on the concentration of the antimitotic agent and the duration of pretreatment, especially when mutagenic chemicals such as colchicine and oryzalin are applied for DNA duplication [18,19]. If successfully achieved, the marked effects of genome doubling could ameliorate unfavorable external pressures such as drought and salinity stress, including deterring pests as well as viral- and bacterial-induced diseases.

According to Corneillie et al. [24], morphometric analysis of polyploidized *Arabidopsis* plants revealed that their epidermal tissues contained increased cell sizes and reduced cell number per mesophyll compared to diploid plants. In line with these observations, the various polyploidized varieties of *Arachis*, *Glycine wightii*, *Medicago sativa*, and *Phaseolus aureus* also demonstrated morphological similarities to *Arabidopsis* polyploids [25]. In view of the abovementioned reports, many studies presented the idea that polyploids offer adaptive advantages to circumvent environmental stress effects than the diploid progenitors, even if their mechanism underlying stress resistance remains elusive. This somewhat holds significant truth since some of the polyploids frequently showed vigorous growth, higher yield, and resistance under stressful conditions than their parent diploid species. Tetraploid plants of *Medicago truncatula* Gaertn and *Medicago scutellate* (L.) Mill recorded up to 100% greater autumn biomass and 58% larger seed weights compared to their diploid parent lines under field conditions [26].

The *M. truncatula* Gaertn and *M. scutellate* (L.) Mill are well known to produce high protein forage content for livestock and a significant fix of atmospheric nitrogen, especially under rainfed farming systems without severe exposure to drought, heat, or salinity stress [27]. However, it is still not yet clear, especially because of the complex regulatory mechanisms involved in plant development and responses to environmental stresses, whether there is a common set of physiological and molecular strategies that polyploidized plants use to mitigate against abiotic stress. Some of the physiological changes that have been observed in polyploidized mutant plants compared to diploid lines, as illustrated in Figure 2 include increased adaptability to environmental stress. Rao et al. [8] and Tossi et al. [28] reported that the positive impact on plant growth and net yield production of crops are due to the diversification of gene functions accompanying this evolutionary process. As expected, chromosome duplication would trigger major physiological changes that lead to notable resistance to stress, including the changing of the morphology of plant organs that confers stress tolerance.

## 3. Physiological and Molecular Status of Leguminous Polyploids

In legumes, polyploidy serves as a major genetic driver in regulating growth and development processes, including the legume-rhizobium symbiosis, which plays an important role in the global nutrient cycle and diversification of plant species, as previously mentioned [29]. Thus, the metaphase inhibition illustrated in Figure 2 may be induced through efficient, reliable, low-cost, and safe pretreatments of plant materials such as cells, organs, or seeds with the different doses of mutagenic chemicals (also demonstrated in Figure 1) for chromosome doubling could result in altered gene transcriptional activities (Figure 2a–c). Accordingly, such interactions may have further consequences that include differential expression of common and distinct gene families depending on the strength and activity of promoter sequences. Some of these expressed genes may encode proteins that are responsible for significant notable physiological and phenotypic changes that enhance the resistance of leguminous plants to abiotic stresses such as salinity, drought, and high temperature. However, at this molecular level, salinity influences the expression of genes related to the biochemical parameters associated with salt tolerance. For instance, in non-leguminous species, such as *Steria rebaudiana* Bertoni (an important medicinal plant native to northeastern Paraguay and southern Brazil), the overexpression of peroxidase encoding genes (*POD4*, *POD6*, and *POD9*) was reported [30].

The expression of these genes virtually influenced all metabolic reactions and physiological processes, specifically the functional pathways, nutritional imbalances, and hormonal regulations, eventually getting all these impeded. Although, unlike the widely reported drought-triggered effects on gene regulation and expression, the effects of salinity are more physiologically pronounced, involving major biochemical metabolite synthesis, enzyme induction, and membrane transport than the upregulation and downregulation of plant genes. If mineral ions (Na^+^, Cl^−^, and K^+^) in the soil are in excess, such conditions will inhibit plant growth by interfering with plant cell osmotic adjustment and then reduce ROS scavenging activity through specialized enzymes and antioxidants [10]. Razzaque et al. [31] reported allelic diversity at several genetic loci that have been found in genotypes that show tolerance to an excess of these salt-stress-inducing ions (Na^+^, Cl^−^ and K^+^), influencing the growth and nutritive value of leguminous crops. However, the marked physiological effects in polyploids because of genome doubling and altered gene functions revealed increased adaptability to environmental variability [32]. Genome duplication may lead to the improvement and/or divergence of an entire duplicated pathway, among others, resulting in noticeable changes in the physiological traits related to water use efficiency and photosynthesis. Such alterations include those of chlorophyll and hydrogen peroxide contents, together with abscisic acid (ABA), which were regarded as physiological indicators to judge the response of polyploidized *L. ruthenicum* plants against environmental stress [8]. Under reduced soil water content, which also led to osmotic stress from dehydration, the diploids showed inhibition of chlorophyll synthesis and an excessive increase in the concentration of hydrogen peroxide (H_2_O_2_). The superoxide dismutase produced H_2_O_2_ gets increased during environmental stress, interacting with thiol-containing proteins and activating the different signaling pathways, including transcriptional factors (TFs) involved in the regulation of gene expression and cell-cycle processes in plants.

## 4. Ionic and Osmotic Stress Signaling Responses in Polyploidized Plants

To cope with salinity stress, plants have developed several adaptive strategies, such as enhanced enzymatic and physiological reactions, as well as higher levels of metabolite expression, such as phenols that are involved in the antioxidant activities of the plant. This stress affects vital metabolic functions whereby Na^+^ and Cl^−^ ions interfere with several nutrients, such as those highlighted in Figure 3. Salt stress affects competitive nutrient uptake, accumulation, and transport in plants [33], consequently disturbing nutritional balance and hormone interactions, causing osmotic stress effects and specific ion toxicity [34]. In soybean, the ratio of gibberellin (GA) and abscisic acid (ABA) was negatively regulated by NaCl during seed germination and post-germinative growth [35]. This report further indicated that NaCl considerably down-regulated GA_1_, GA_3_, and GA_4_, while regulating the ABA content. As outlined in Figure 3, high soil salt content affects crops’ growth and development processes by influencing water availability in the cells, ion toxicity, oxidative stress, cell membrane organization, meiotic/mitotic processes, and nutritional disorders, among others.

In terms of inorganic ions, essential elements (N, P, K, S, Mg, etc.) are considered intrinsic components of the structure and metabolism in plants [6,39]. The severe abnormalities that may arise due to salinity stress will result in symptoms of specific nutritional deficiencies such as chlorosis and necrosis. It remains common knowledge that salt stress influences the transfer of inorganic nutrients from the environment into a plant [40], but further insights are still required on how nonspecific osmotic stress and specific ion effects are regulated at both physiological and molecular level. Even though, the mechanisms through which legumes with altered ploidy adapted to salinity stress still need to be investigated, insights on other non-leguminous crops demonstrated novel traits and changes in existing physiological processes because of polyploidization.

As previously reported, polyploidized mutants always showed improved biosynthetic, carbolic, and signaling transduction pathways under environmental stress, as indicated in Figure 2 and Figure 3. In the general scheme of signal transduction, polyploidized plants would likely show rapid environmental and developmental signal perception by specialized receptors and effective activation of a signaling cascade involving secondary messengers, leading to an efficient response by the plant cells (Figure 3) [31,38]. Part of polyploids’ success is signaling between plants and microbes during symbiotic nitrogen fixation (SNF), which influences microbial infection of plant cells, cell division leading to nodulation, autoregulation of nodule development, and intracellular accommodation of bacteria, metabolism, and transport supporting SNF to prevent nodule senescence [41].

Salinity stress increases the accumulation of reactive oxygen species (ROS); however, total phenolic content, ascorbic acid content, and antioxidant activity traits were enhanced in 320 tetraploids compared to 84 diploid genotypes of *Solanum tuberosum* L. [42]. According to the study’s revelations, the concentration of antioxidant compounds correlated positively with the skin tuber color and ploidy levels. In *Citrus limon* (L.) Osbeck, Bhuvaneswari et al. [43] indicated that cyclic monoterpene, limonene, had significantly increased in tetraploids induced by 0.025% colchicine for 24 h. Limonene is biosynthesized from the precursor geranyl diphosphate (GPP) by enzymatic biotransformation with d-or-l-limonene synthetase. The production of limonene monoterpenes is completed by methyl group deprotonation in the α-terpinyl cation, and this occurs in a wide variety of species, including Conifers, Lamiales, and Pinales, of which some are tetraploids [44,45].

Moreover, overexpression of the terpenoid biosynthesis gene in soybeans influenced the nodulation signaling pathways. The results indicated that the six terpenoid synthesis genes (*SoTPS6*, *SoNEOD*, *SoLINS*, *SoSABS*, *SoGPS*, and *SoCINS*) in soybean hairy roots increased nodule number, nodule root length, and fresh root weight [46]. As reported in other studies, the expression of terpene in the roots was highly coordinated and cell-specific. The abovementioned observations demonstrated the role of ploidy in this largest class of specialized metabolites found within different structures in plants. According to Pott et al. [47], monoterpenes like limonene plays a key role in plant’s interaction with the environment, especially in protecting plants against biotic and abiotic stresses.

Generally, polyploid plants often have improved traits than their diploid relatives and may have a greater advantage in regulating both osmotic and ionic stress. The improved traits or maintained homeostasis plays an important role in many of the functions occurring in polyploidized plant cells. The accepted explanation of the influence of salinity stress is that imbalances in ion content in plant cells affect plant fitness, especially by affecting plant nutritional status and cell water potential, and cause toxic effects through accumulation of ions, which disturb nutrient acquisition and result in cytotoxicity [48,49]. Under stress, polyploidized plants show increased capacity to maintain unequal concentrations of sodium (Na^+^) and potassium (K^+^) ions both inside and outside the cells. Such improved responses occur due to tolerance genes and TFs regulating ion Na^+^/K^+^ pump and exclusion through selective permeability of plasma membrane proteins (PMP), high sodium affinity transporter (HKT), and Na^+^/H^+^ exchangers that help to alleviate ion toxicity in the cells [50,51].

Although information regarding polyploidized leguminous crops’ reaction to toxic ions is very scant, monocotyledous crops such as hexaploidy wheat spp. (*Triticum aestivum* genome BBAADD) demonstrated improved ion toxicity alleviation than both diploid counterparts and tetraploid wheat progenitors (*T. turgidum*) or durum wheat (*T. durum*). In polyploidized rice (HN2026-4x and Nipponbare-4x), Tu et al. [52] reported improved regulation of Na^+^ content and H^+^ proton content flux at root tips, with decreased Na^+^ and increased H^+^ efflux in the roots. In legumes, the physiological mechanisms explaining the improvement in salt tolerance with increasing levels of ploidy remain under-researched. However, genome duplication has been widely suggested to improve salinity stress resistance through enhanced proton transport pump and homeostasis in various monocot and dicot vegetable crop species [51,52,53].

## 5. Metabolic Changes of Legume Polyploids Exposed to Salinity Stress

Salinity stress limits crop growth and metabolism, posing major impacts on agricultural productivity. As previously mentioned, over-irrigation systems and drought stress continue to cause oversalinization in many areas used for planting agroeconomically important crops. Salinity stress, also known as hyperionization, poses serious effects on the metabolic and growth processes of plants through excessive accumulation of ions, as previously indicated [54]. Numerous reports have elaborated on NaCl’s effects on growth, mineral composition, proline content, and antioxidant enzyme activity, which dramatically reduces the productivity of many crops. As highlighted in Figure 3, this negatively influences energy production, utilization, and storage in plants, generating ROS, reducing cell growths, assimilating production, and causing membrane dysfunctions [55], consequently disrupting growth, reproduction, and yield in many crop species. Hamayun et al. [56] reported significant decreases in endogenous levels of secondary metabolites, GA, jasmonic acid, salicylic acid (SA) and ABA due to salinity stress. However, abrupt changes in the osmotic and ionic state of the plant cause negative effects on the crop’s metabolism and growth.

These depressive effects of NaCl were also demonstrated in a previous study conducted in our laboratory (Figure 4), where diploid plants showed inhibition in soybean growth and yield characteristics [57]. The soybean plants were subjected to two levels of salinity stress. Similar findings were earlier reported by Amirjani [58] and Hamayun et al. [56] in this species, cultivar Hwangkeumkong. The observations made in Figure 4a indicated that soybean plants exhibited noticeable morphological variations, particularly in their root and stomatal density, compared to the marginal but significant reductions demonstrated in shoot length, number of branches, and leaf area. The salt-stressed plants clearly indicated that yield components in both cultivars were decisively determined by the stress (Figure 4a), especially flowering and the number of pods produced per plant. As a consequence of the stress, lower flowering percentages and pod numbers were recorded [57]. In contrast, mutant plants gave improved results in flowering and the number of pods per plant using 0.8% and 0.6% EMS, respectively, in mungbean (*Vigna radiata* L.), according to Roychowdhury et al. [59].

Additionally, the mean number of seeds per pod and 100-seed weight were increased in all the mutagenic treatments involving gamma rays and EMS in mungbean, urdbean varieties T-9 and Pant U-30 [60]. These mutation-based improvements conferred increased genetic diversity required to achieve economically important traits in both leguminous and non-leguminous crops wherein the selected mutant lines successfully contributed to the diversity of the crop’s genetic base, which is highly required for growth improvement against environmental stresses such as drought and salinity. Furthermore, the genetic variability induced through mutagenesis in treated populations also demonstrated some level of genotype dependency, as reported by Laskar et al. [61] in lentils (*Lens culinaris* Medik). The genome structure in polyploids also has implications for both physiology and stress responses during vegetative and reproductive growths. As reports showed that polyploidy has occurred at least once in angiosperm lineages, the significantly altered phenotypes often influence the interactions of polyploids towards both biotic and abiotic stress constraints. According to Forrester and Ashman [29], plant-biotic interaction involving legume-rhizobia mutualism serves as an important interaction that regulates the nutrient cycle, in addition to its indirect impact on vegetative (number of trifoliate leaves, leaf size, shoot height, root length, branch number, etc.) and reproductive growth parameters (flower number, fruit pods, seed number, and 100-seed weight) as indicated in Figure 4a,b.

Although some researchers found that the notion that autopolyploids are larger than their diploid progenitors does not always hold true, such as the comparison of *Arabidopsis* and *Musa* plants, wherein obtained variations brought from haploidy to diploidy showed increases in growth parameters but then decreased with the increase in ploidy levels in both species. Such findings were reported by Corneillie et al. [24] in *A. thaliana* and Brisibe and Ekanem [62] in *Musa* species. Quantitatively and qualitatively studying these differences in mutant plants can allow scientists to infer the functions of duplicated genes or map certain mutations on the chromosomes. Furthermore, such studies must continue to selectively breed crops to produce varieties that have higher yields or varieties adapted to specific environmental conditions and are resistant to plant pathogens. As it is apparent from Figure 4, polyploidization could substantially improve the tolerance of legumes to salinity stress, even though this may result in losses of certain important characters. Leguminous crops such as soybean, chickpea, dry bean, cowpea, lentil, and peas are important sources of proteins and oils, so their potential improvement via ploidy duplication must rather be associated with the improvement in these useful economic traits.

## 6. Gene Expression and Symbiosis in Polyploids Grown under Salinity Conditions

Although a detailed understanding of the molecular and physiological mechanisms explaining the improvement in salinity stress tolerance with increasing levels of ploidy in leguminous crops remains scant. Tossi et al. [28] reported the change in some leaf traits that favored abiotic stress tolerance in *Arachis duranensis* × *A. ipaensis* synthetic allotetraploids. In urdbean (*Vigna mungo* (L.) Hepper), mutagenized genetic variability contributed to improved phenotypic traits in M_2_ and M_3_ generations [60]. Chao et al. [63] earlier reported improved salinity stress resistance in autopolyplodized *Arabidopsis thaliana* through enhanced potassium (K) accumulation in root and mesophyll tissues. The study revealed that ploidy level was a significant determinant of leaf K concentration attributed to the 57.2% variations observed in tetraploids, constituting a 32% K-level increase in these plants compared to the diploid lines. Furthermore, the findings showed that autotetraploid accession Wa-1 (Warsaw, Poland) contributed additional alleles for increased leaf K with no obvious traits found in their diploid counterparts. According to Munns and Tester [64], increased K/Na ratio enhanced tolerance to osmotic and ionic components of salinity stress, as similarly reported by Chao et al. [63].

Isayenkov and Maathuis [65] also reported the adaptation to adverse environmental salinity stress conditions in polyploidized plants driven by genes related to the efficient operation of NADPH-dependent ‘ROS-Ca^2+^ hub’. This multigenic family of calcium-dependent protein kinase encodes structurally conserved unimolecular calcium sensors or protein kinase effector proteins [66]. As reported in previous studies, Ca^2+^-dependent protein kinase diversity could be amplified by splicing and post-translational modifications [67], with a proposition that these protein kinases follow a common regulatory mechanism through phosphorylation [68] to confer tolerance to salinity stress. Hexaploidy lines of *Ipomoea trifida* reportedly exhibited increased K^+^ retention while excluding Na^+^, presenting highly reduced sensitivity of plasma membrane K^+^-permeability channels in both mature and elongation root zones of plants [69]. The abovementioned ion channel activation occurs as a result of ROS-induced protein modification/membrane-dependent NADPH oxidase system, regulated by the expression of respiratory burst oxidase homolog (*Rboh*) genes [70,71]. The retainment of high K^+^ levels and exclusion/accumulation of low Na^+^ concentrations in the cytosol confers salinity stress tolerance in plants [69].

In many legumes, specific redox-dependent signaling pathways involving ROS, reactive nitrogen species (RNS), and reactive sulfur species (RSS) play a key role in acclimation and tolerance to environmental stress. According to Matamoros and Becana [72], ROS-induced modifications in legumes occur through oxidation, S-nitrosylasion, S-glutathionylation or per sulfidation of the cysteine thiol group, oxidation of methionine residue, nitration of tyrosine residue and lysine/arginine residue carbonylation. Molecular analysis of redox-based post-translational modification of proteins revealed functional implications of these alterations on nodulation and whole plant growth and development under adverse conditions. Even though polyploidy is a major genetic driver of ecological and evolutional diversity in plants, its effects on legume interactions remain partially explored on the mutualistic relationship of legumes with microbes. Such studies, like that of Forrester and Ashman [25], reported that *Medicago sativa* subsp. *caerulea* autotetraploid plants produced larger nodules with larger nitrogen fixation zones than diploid plants using two strains of rhizobia, *Sinorhizobium meiloti* and *Sinorhizobium medica*. Focal microscopy was used to quantify the internal traits of nodule formation in both diploid and neotetraploid *M. sativa* plants.

There was a strong direct effect on nodulation and nodule traits such as N-fixation zone, nodule area, and bacteroids containing symbiosomes due to increased ploidy level. These nodule features and other polyploidy-induced unique characteristics involving activation of the nodule-specific transcriptome, node genes within mobile plasmids of rhizobial genomes, and ample nodule-specific cysteine-rich (*NRC*) genes (Table 1) were reported. *NCR* gene family is extensively large in *Medicago* spp. with about 600 other genes [73,74]. The wider availability of such genome sequence information of *Medicago* and several other legume species will boost genomic research and breeding for yield improvement and adaptation of plants to adverse conditions. RNA sequencing transcriptome analyses in polyploids provided information into the transcriptional mechanisms, molecular mediation of various cellular processes, genome reprogramming, and gene function, as indicated in Table 1. As far as a decade ago, transcriptome analysis has been performed in several diploid legumes such as *Glycine max* (L.) Merrill, *M*. *truncatula* Guertn, *Lotus corniculatus* L. var. *japonicus* Ragel and *Cicer arietinum* L. to unravel the overall and specific transcriptional activities of genes in both diploid and polyploidized lines [75]. In shoots and roots of *M*. *truncatula*, transcriptional responses to treatment with rhizobia detected varied gene expression pattern shifts under different conditions [76].

Some of the genes, like leginsulin, defensins, root transporters, nodule-related genes, and circadian clock genes, are widely explored across legume species in relation to biological symbiotic nodulation systems (Table 1) [88,89,90]. Currently, sufficient information is available in legumes that are required to elucidate responsible genes and cellular processes involved in plant-microbe symbiosis; however, many genes in the nodulation, plant growth, and response to abiotic stress, such as salt and drought stress still need to be investigated. Furthermore, not all genes are upregulated during symbiosis or the period of stress. Genes such as *MtDef4.3* (Table 1) of plant defensin genes in *M. truncatula* be downregulated under rhizobia-induced nodulation for N-fixation [76]. Most of these expressed genes contribute to the morphology, physiology, and developmental attributes, also providing housekeeping functions necessary to normalize the expression, regulation, and response of plants to environmental instabilities.

The *WRKY* genes encoding WRKY proteins containing conserved sequence WRKYGQK heptapeptide at the N-terminal end have also been widely implicated in leguminous and non-leguminous crop tolerance to biotic and abiotic stress. Although WRKY evolution in legumes is still unknown, Song et al. [91] reported that this family comprises a class of TFs involved in physiological changes that enhance plant responses under abiotic stress, such as drought and salinity constraints. As alluded to by various other researchers, natural and induced polyploidization via mutagenesis can unleash new alleles of WRKY genes and others that control traits required for salinity stress tolerance.

## 7. Discussion and Prospects

Salinity stress has caused adverse negative impacts on plants for decades, limiting the growth, reproduction, and survival of many plant species. The kinds of pressures inflicted by this stress during plant growth and development include osmotic stress, ion toxicity, and reduced cell water potential, which constitutes water deficit stress. According to Zhao et al. [92], salt stress severely affects plants throughout their life cycle, hindering seed germination, growth and development, flowering, and fruiting. In addition to its toxicity, salt stress limits nutrient uptake by roots. Stress-induced Na^+^ ions also function as signaling molecules of this abiotic stress and drought, especially after perception by non-selective cation channels [31]. Non-selective cation channels are multi-protein complexes found within cellular membranes, and they give high permeability to ions, mainly Na^+^, K^+^, Ca^2+^, and Cl^−^, through polarization of the plasma membranes [93]. In cowpeas, Na^+^, K^+^, K/Na ratio, seedling heights, and chlorophyll were significantly influenced by salt stress [94]. Mungbean plants also exposed to saline conditions exhibited reduced growth and yield were unable to take up sufficient water required for metabolic processes, and failed to maintain turgidity due to low osmotic potential in the cells [95]. Other leguminous crops such as soybean, lupin, pea, peanut, and pigeon peas are also seriously affected by salinity stress; however, their intrinsic tolerance mechanism and agro-physiological characteristics still need to be explored.

As alluded by HanumanthaRao et al. [95], salinity tolerance comprises multifaceted responses at the molecular and physiological levels, but much remains to be studied from the natural and adaptive tolerance to this stress in both diploid and polyploidized leguminous plants. It is evident that ploidy level, particularly in leguminous crops that are widely studied/serving as industrial crops and essential to ensure food security, is poorly correlated with salinity stress and any other abiotic stress resistance. This poor correlation may be due to the overwhelming number of duplicated sets of genes that are involved at different developmental stages in coping with salt stress. It is, therefore, important to comparatively evaluate diploid and polyploid stress tolerance traits and further explore these traits in breeding programs to obtain new lines for durable tolerance. As HanumanthaRao et al. [95] reported, tolerance in salinity stress is achievable in legumes via the overexpression of housekeeping differentially expressed genes such as *SOS1* involved in the exclusion of toxic Na^+^ ions by plasma membrane Na^+^/H^+^ antiporter, sequestration of Na^+^ into the vacuoles to reduce ion toxicity and encoding compatible osmolytes to protect cell proteins against salinity-induced inactivation and denaturation. Such osmolytes, predominantly amino acids, amino acid derivatives, sugars, and polyols, as reported by Kurz [96], are synthesized or taken up by plants for adaptation purposes under extreme environmental conditions.

Generally, during abiotic stress, osmolytes encoded by the raffinose family oligosaccharides (*RFO*) genes produce enzymes that convert sucrose into RFOs, which establish cell stability and offer protection against abiotic stress such as salinity, drought, and cold stress [97]. These candidate *RFO* genes were quantitatively analyzed in *Phaseolus vulgaris* and *Glycine max* by the above report, revealing that reduction in RFO content in the seeds could improve the nutritional quality of these beans without compromising normal plant developmental and functional characteristics, especially under abiotic stress conditions. The abovementioned intrinsic mechanism of stress response in plants exhibits enhanced stress tolerance in polyploid plants relative to their diploid counterparts. Wei et al. [98] reported that autotetraploid trifoliate orange (*Poncirus trifoliata* (L.) Raf.) accumulated higher levels of soluble osmolyte sugars and proline under salt stress compared to the diploids. In addition, polyploids exhibited enhanced salt tolerance by displaying specific enrichment of *DEG* genes with varying gene expression patterns in both roots and shoots. As indicated in many other studies, genes related to hormone signal transduction, starch and sucrose metabolism, proline biosynthesis, antioxidant enzyme activity, and several specific genes encoding transcriptional factors potentially being over-expressed in polyploid plants [10,20,32,43,71].

## 8. Conclusions

Studying the response of leguminous crops under different stresses and environments at the physiological and molecular levels and comparing the gathered information between diploids and polyploids will enable us to understand and measure the beneficial activities of the majority of genes found in these plants. This will also provide further insights into gene regulation under different biological contexts, identify genes that are important for the various plant growth and developmental processes, and strengthen in-depth investigations and physio-molecular understanding underlying salinity stress tolerance in leguminous crops. Understanding the physiological and molecular responses of polyploidy-induced tolerance to abiotic constraints such as salinity and drought stress could provide correlative insights into legume biology and enhance our understanding and acceleration towards the establishment of efficacious breeding approaches. As succinctly reviewed in this paper, polyploidy remains the driving force in the evolutionary adaptation and success of leguminous crops, thus artificially induced autopolyploidy using chemical agents such as colchicine and oryzalin generated anatomical, morphological, physiological, and molecular changes increasing tolerance to abiotic and biotic stresses, which positively impact of plant growth and yield productivity.

## Figures and Tables

**Figure 1 cells-12-02082-f001:**
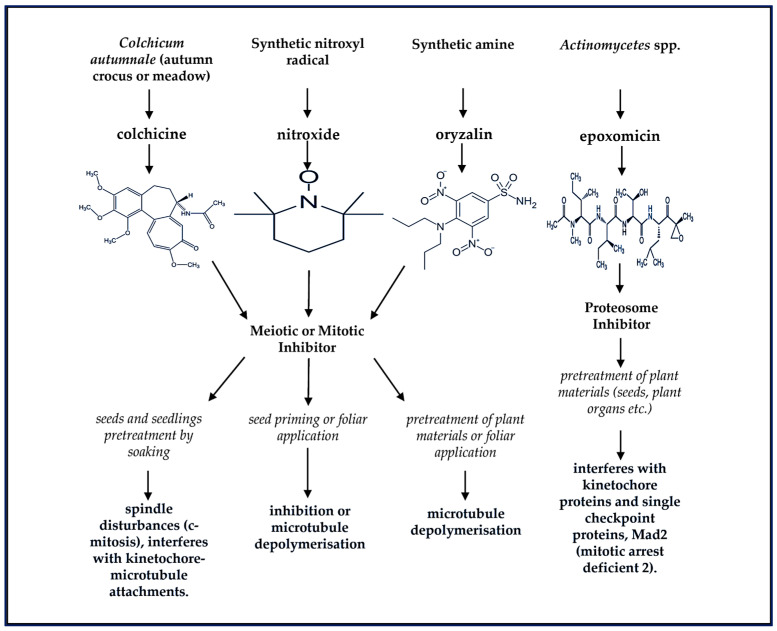
Chemical structures of naturally and synthetically produced mutagenic compounds are used to achieve the multiplication of complete sets of chromosomes and the production of random mutations in the genetic materials of plant species, giving rise to the formation of new genetic variations. These multiple sets of chromosomes and nucleotide alterations (particularly, the substitution of guanine (G)- cytosine (C) to adenine (A) alkylation [G:C to A] induced by ethyl methane sulfonate) may take place in one nucleus and can be stably inherited to progenies [12,14,16].

**Figure 2 cells-12-02082-f002:**
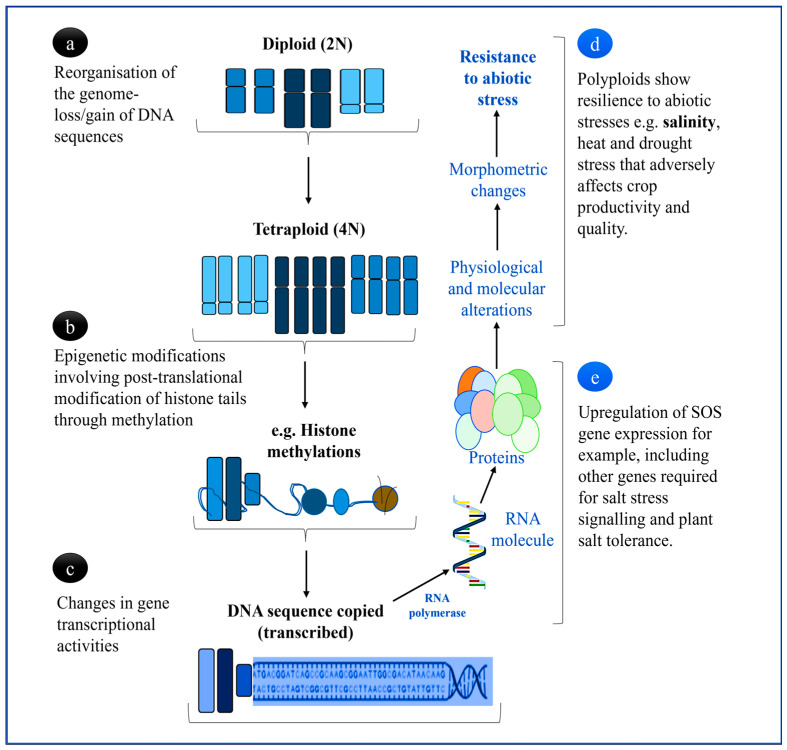
Summary of polyploidization events of the genome in plants and their role in abiotic stress tolerance enhanced by chromosome duplication. (**a**) Genome reorganization facilitating increased allo- or auto-polyploidization, either leading to extensive gene duplication or fractionation (loss) following polyploidy induction, (**b**) DNA modifications that regulate whether genes are turned on or off after polyploidization, reprogramming gene expression networks, and (**c**) ploidy-related modifications on gene structure and function as a result of the depolymerization of microtubule activities causing great alteration in the expression patterns of RNA-mediated polypeptides (**d**,**e**), which are also influenced by external stimuli such as environmental stress (**d**) [18,21,23].

**Figure 3 cells-12-02082-f003:**
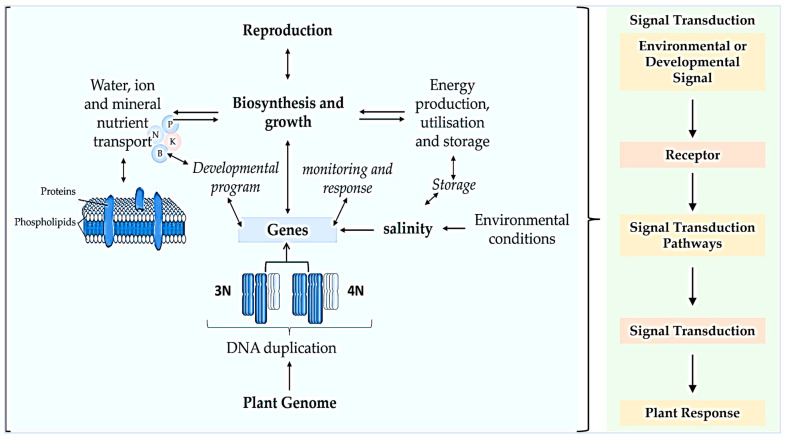
Biochemical and physiological events in the sensing and response of plants to salinity stress. The prevalence and recurrence of polyploidization in plant species make it one of the most important evolutionary events [36,37] in the diversification of species through DNA duplication, altered developmental and monitoring responses, energy production, utilization, and storage during abiotic stress and evolution of genes, as well as high reproduction. Environmental or developmental signals include light, temperature, nutrients, etc.; receptors such as kinases, ion channels and G-protein-coupled-receptors; repressor protein degradation and protein phosphorylation; hormonal transport, etc.; and responses by transcriptional/gene expression and post-translational modifications were improved in polyploidized plants [31,38].

**Figure 4 cells-12-02082-f004:**
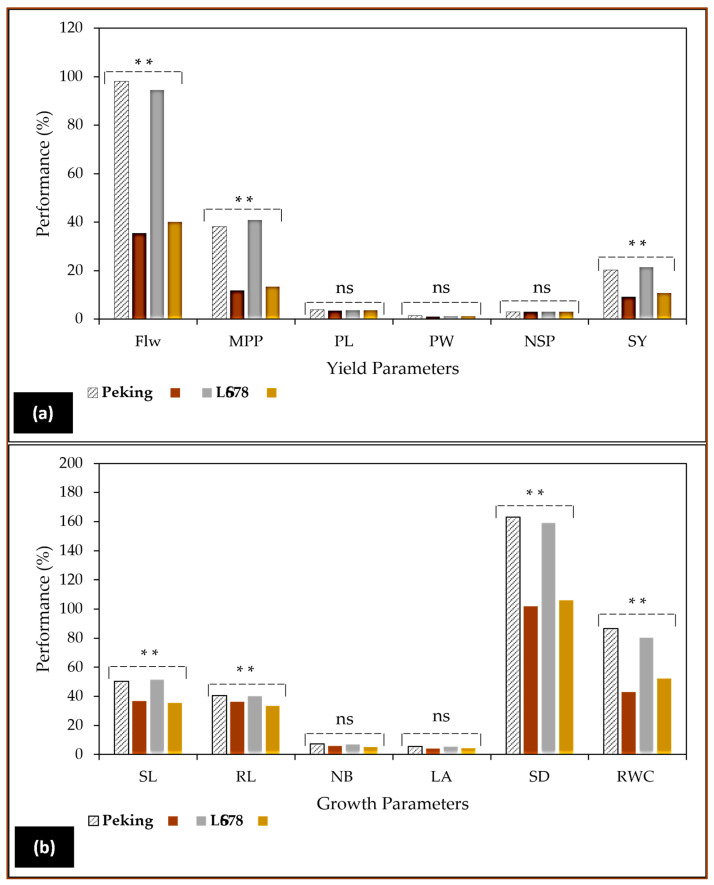
The changes on reproductive yield (**a**) and vegetative (V) growth (**b**) caused by the increase in substrate salinity (NaCl) in soybean (*Glycine max* L. Merr.), cultivar LS678 and Peking [57]. Soybean plants were cultivated in 25 cm plastic pots containing pre-sterilized vermiculite and then irrigated weekly with 250 mM NaCl from V2 until V3 stage under in vivo conditions. This study revealed that soybean plants subjected to salinity stress exhibited highly reduced growth [shoot length (SL), root length (RL), number of branches (NB), leaf area (LA), stomatal density (SD) and relative water content RWC)] and yield parameters [flower number (Flw), mean pods per plant (MPP), pod length (PL), pod weight (PW), number of seeds per pod (NSP) and seed yield (SY)], in addition to biochemical features such as photosynthetic pigments, carbohydrates, phenolic content, flavonoids, and antioxidant capacity as reported in this cited report [57]. ** Significant at 0.05 probability level, and ^ns^ not significantly different at 0.05 probability level.

**Table 1 cells-12-02082-t001:** Candidate genes conferring salinity stress tolerance in leguminous crops.

Common Name	Species Name	Gene Family/Name	Function	Reference
Peanut	*Arachis hypogeal* L.	*ABSCISIC ACID INSENSITITVE 4S* (*AhAB14s*)	Gene downregulation enhanced survival rate, biomass accumulation, and root/shoot ratio of seedlings	Luo et al. [77]
Chickpea	*Cicer arietinum*	CaRab-GTP (*CaRabA2*, *CaRab-B*, *CabRab-C*, *CabRab-D*, *Cab-Rab-E*, and *CabRab-H*)	Regulate Na^+^ accumulation in leaves	Sweetman et al. [38]
Pigeon pea	*Cajanus cajan* L.	*GAPDH*, *UBC* and *HSP90*	Housekeeping genes carrying out cellular maintenance through stable expression irrespective of internal/external signals	Sinha et al. [78]
Soybean	*Glycine max* L. Merrill.	*GmSALT3* and *GmSALT18*	Regulate Na^+^ accumulation in leaf mesophylls via root-based mechanisms	Guo et al. [79]
Birdsfoot trefoil	*Lotus japonicus*	*LjLTP10*	Gene preventing osmotic stress by managing water deficit through structural architecture such as cuticular composition.	Tapia et al. [80]
Barrel medic	*Medicago truncatula*	*MtDef4.3*	Defense gene against infectious microbes, including rhizobia species.	Gao et al. [76]
Lupin	*Lipinus angustifolius*	Aldehyde dehydrogenase (*ALDH*)	Catalyse irreversible oxidation of aldehyde molecules to non-toxic carboxylic acids	Jimencz-Lopez [81]
Common bean	*Phaseolus vulgaris*	*PvDREB*	Circumvent salinity stress by mechanic and osmotic adjustment	Konzen et al. [82]
Barrel medic	*M. truncatula*	Nodule-specific Cysteine Rich (*NCR*)	Control nodule organogenesis and gene duplication.	Guefrachi et al. [73]
Cowpea	*Vigna unguiculata* ssp. *sesquipedalis* (asparagus bean)	Root-derived DEGs (*GmsSOS1*, *SOS2* and *OsNHX2*)	Genes involved in redox reactions, enhancing antioxidant enzymes activity and reduces Na+ accumulation under salt stress	Pan et al. [83]
Mung bean	*Vigna radiata* L. Wilczek	*VrKUP*, *VrHAK* and *VrKT*	Regulate influx and efflux of K^+^ through K^+^ transporters and channels	Azzam et al. [3], Ragel et al. [84]
Red clover	*Trifolium* spp. (*T. pratense* L.)	*NHX1*	Na^+^/H^+^ antiporter in plants, predominantly, *Trifolium* spp.	Delormel [85]
Pea	*Pisum sativum*	*PsRPL3OE*, *PsChla/bBP* and *PsFDH*	Regulate protein synthesis, photosynthesis, and long-chain lipid synthesis in the mesophylls	Joshi et al. [86]
Wild peanut	*Arachis duranensis*	AdWRKYs (*AdWRKY18*, *AdWRKY40*, *AdWRKY42*, *AdWRKY56* and *AdWRKY64*)	Genes regulating ABA accumulation, stomatal aperture, and root development	Zhang et al. [87]

## Data Availability

Not applicable.

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
