# Peer review of "Cell Mutagenic Autopolyploidy Enhances Salinity Stress Tolerance in Leguminous Crops"

_cells, 2023, doi:10.3390/cells12162082_

Round 1

Reviewer 1 Report

This review paper aims to summarize the current knowledge of the impact of autopolyploidy on salinity tolerance in leguminous crops. While the causal link between the ploidy level and abiotic stress tolerance is well known, specific details of this process remain elusive. In this context, the topic of this review is highly timely.

Being generally positive about this approach, I was disappointed by the actual content and the quality of writing. My major concerns are as follows:

1    Plant adaptation to salinity requires them to handle at least three major stress components – namely, water deficit; ionic imbalance; and oxidative stress (Zhao et al. 2020 The Innovation). This relies on numerous traits such as efficient Na exclusion from uptake, its transport and sequestration in vacuoles; cytosolic K+ retention; osmotic adjustment; higher water use efficiency; and ROS detoxification. Plants also adapt to salinity via set of specific morphological and anatomical alterations (e.g., control of stomata development; leaf succulence; etc). Most of these topics were simply not covered in this review, and the impact of polyploidy on these traits was barely mentioned (if any).

2.       The writing style is appalling and need improvement. The paper lacks critical analysis of the data and is written in a style “Rao et al reported….”, “According to Forrester and Ashman…”, “indicated by Gupta and Huang…” …”. Much better integration is needed.

3.        The paper is full of specific (and unnecessary) details. A good example is ln 320-321: “lower flowering percentages and pod numbers were recorded for LS678 and Peking at 40.1%, 13.4 and 35.5%, 11.8, respectively [57]”. Why is it important to give names of specific cultivars? And what is the need for such accuracy? (e.g., 40.1 %)

4.       The paper contains many statements that are either untrue or too generalized. A good example is in ln 55-56, where the author states that “molecular, and physiological variations between polyploids and diploids were mainly related to the biosynthesis and signal transduction pathways of ABA”. This is not true. For example, papers by Liu et al. (2019) JXBot and Shabala (2019) JXBot both deal with operation of NADPH-dependent ‘ROS–Ca2+ hub’ and regulation of Ca2+-permeable channels by ROS in plant adaptive responses to salinity. Another example is in ln 76-78: “As reported by Farooq et al [11], unravelling the response mechanisms and tolerance…. may help in understanding the performance ...”. To start with, this is a review paper that cannot “report”. Second, this statement is trivial and was made in hundreds of previously published papers. Why this misleading citation then?  

adequate

Author Response

RESPONSE TO REVIEWERS’ COMMENTS

Journal: Cells

Manuscript number: cells-2488523

Manuscript title: Cell mutagenic autopolyploidy enhances salinity stress tolerance in leguminous crops

REVIEWER 1

Comment 1: Plant adaptation to salinity requires them to handle at least three major stress components- namely, water deficit, ionic imbalance and oxidative stress (Zhao et al. 2020 The Innovation). This relies on numerous traits such as efficient Na exclusion from uptake, its transport and sequestration in vacuoles, cytosolic K+ retention, osmotic adjustment, higher water use efficiency and ROS detoxification.

Reply 1: Thank you for the comment. Please note that this was thoroughly considered and attended to accordingly. In preparing this manuscript we mention effects of salt stress without/avoiding repeating what is already widely reported in the literature. These areas are highlighted in different sections of the manuscript (e.g., see section 4 Line 217-231), which addresses concerns raised by the reviewer. Potential response of polyploids to oxidative stress through antioxidant activity or ion exclusions are also covered in the same section Line 264-274 and then on 287-296, 479-483, 500-508. As indicated very limited reports are available on this topic, thus, the specific analysis of the role of ploidy on these grain crops under saline conditions would be impossible to argue currently in this paper.  

Furthermore, unlike transgenic plants, autopolyploids do not add new foreign genes but rather causes duplication of already available genes, coming from one single species.

Comment 2: Plants also adapt to salinity via set of specific morphological and anatomical alterations (e.g., control stomata development; leaf succulence etc.). Most of these topics were simply not covered in this review, and the impact of polyploidy on these traits was barely mentioned (if any).

Reply 2: Thank you for the comment. Please note that this was attended to accordingly. The purpose of this review focused on the molecular and physiological influence of polyploidy on legume crop response to salinity stress. We recently published a paper in the MDPI journal PLANTS that covered this aspect in detail (https://doi.org/10.3390/plants12061356) and thus, there was a great need to avoid repetition of the previous work. Nevertheless, such morphological changes were briefly mentioned in section 2 (Line 127-134 & 151-153) and others (e.g., Line 328-337).

Comment 3: The writing style is appalling and need improvement. The paper lacks critical analysis of the data and is written in a style “Rao et al reported….”, “According to Forrester and Ashman…”, “indicated by Gupta and Huang…”. Much better integration is needed.

Reply 3: Thank you for the comment and it was attended to accordingly. Changes were made (as highlighted) to improve the manuscript quality as suggested. This paper seeks to provide an updated survey of recently published information on legume ploidy, focusing on major crops’ molecular and physiological aspects, of which such reports are very minimal to none. This is one of the main reasons why this paper was written. However, we do not agree with this comment because a concise and precise review of literature on this topic was required.

Comment 4: The paper is full of specific and unnecessary details. A good example is in Line 320-321. Lower flowering percentages and pod numbers were recorded for LS678 and Peking at 40.1%, 13.4 and 35.5%, 11.8, respectively [57]. Why is it important to give names of specific cultivars? And what is the need for such accuracy? (e.g., 40.1%).

Reply 4: Thank you for the comment and it was attended to accordingly. The sentences were modified accordingly. But please note that this comment was very much confusing as the accuracy and precision are very important factors in science research and scientific writing.

Comment 5:

“The paper contains many statements that are either untrue or too generalized. A good example is in Line 55-56, where the author states that “molecular, and physiological variations between polyploids and diploids were mainly related to the biosynthesis and signal transduction pathways of ABA”

Reply 5: Thank you for the comment and it was attended to accordingly. This has been clarified, the sentence was a continuation from the previous one and was modified as required where duplicated genes responsible for ABA biosynthesis and regulation were upregulated in polyploidised Lycium than its diploid counterparts.

Reviewer 2 Report

General comments:

In this manuscript, authors discusses the physiological and molecular response of legumes towards salinity stress based osmotic and ionic imbalances in plant cells. They describes on a multigenic response involving various compatible solutes, osmolytes, ROS, polyamines and antioxidant activity, and they also mention the genes encoding proteins involved in the signal transduction, regulation and response mechanisms to this stress. Based on these data, they are arguing that polyploidisation is the driving force in plant evolution and adaptation to environmental stress constraints such as drought, feverish temperatures, and in particular, salt stress.

This review argues in its own logic how leguminous crops have acquired the current form of salt stress tolerance mechanism through repeated cell mutagenic autopolyploidy for a long time. Review is well written and information is useful to general public interested in environmental stress response in plants. However, this Ms could be improved with little changes.

Major comments:

1. In pages 2 and 3, authors well describes on roles of autopolyploidy in plant genetic and physiological events:

However, the authors mainly describe polyploidization by artificial induction. If they briefly explain the polyploidization that naturally occurred in leguminous crops and the resulting genetic and physiological variations, it will be more helpful for readers to understand natural phenomena. In particular, please mention the case of events for salt stress resistance in line with the topic of this paper.

2. In page 4, lines 147 and 151, authors write:

 " Figure 1. Chemical structures of naturally and synthetically produced mutagenic compounds that are used to achieve the multiplication of complete sets of chromosomes and production of random mutations in the genetic materials of plant species, giving rise to the birth of new varieties." According to this caption, it is understood that new plant varieties are generated through random mutations and multiplication of complete sets of chromosomes by natural or synthesized mutagenic substances. If so, it is necessary to tabulate and explain the examples of new plant varieties in the text by organizing them by natural or synthetically produced mutagenic compounds that have induced random mutations and the polyploidization of chromosome sets.

3. In pages 12 and 13, “Discussion and Prospects”:

To add confidence to the authors' arguments, it should be discussed in depth how genetic variations induced through random mutations by naturally and synthetically produced mutagenic compounds and the polyploidy of chromosome sets in leguminous crops can be maintained stably for the next generations.

Considering the wide readership, please clarify the content of the text by using scientific but general and easy terms.

Please revise the text to a more concise sentence.

Author Response

RESPONSE TO REVIEWERS’ COMMENTS

Journal: Cells

Manuscript number: cells-2488523

Manuscript title: Cell mutagenic autopolyploidy enhances salinity stress tolerance in leguminous crops

REVIEWER 2

Comment 1: “In pages 2 and 3, authors well describe on roles of autopolyploidy in plant genetic and physiological events, However, the authors mainly describe polyploidisation by artificial induction. If they briefly explain the polyploidisation that naturally occurred in leguminous crops and the resulting genetic and physiological variations, it will be more helpful for readers to understand natural phenomena. In particular, please mention the case of events for salt stress resistance in line with the topic of this paper.“

Reply 1: Thank you for the comment and it was attended to accordingly. The natural form of polyploidization was briefly described in paragraph 2, Line 50-55, and also in Line 39-41. This was also clarified under section 2 (Line 104-113), first paragraph. Meanwhile evidence of allopolyploids and autopolyploids’ resistance to salt stress are indicated in section 2 (Line 127-131) whereby wild forage legumes are discussed instead of domesticated grain legumes due to the lack of published information. Please note that further non-legume examples are thoroughly highlighted in section 4.

Comment 2: In page 4, lines 147 and 151, authors write: “Figure 1. Chemical structures of naturally and synthetically produced mutagenic compounds that are used to achieve the multiplication of complete sets of chromosomes and production of random mutations in the genetic materials of plant species, giving rise to the birth of new varieties.” According to this caption, it is understood that new plant varieties are generated through random mutations and multiplication of complete sets of chromosomes by natural or synthesized mutagenic substances. If so, it is necessary to tabulate and explain the examples of new plant varieties in the text by organizing them by natural or synthetically produced mutagenic compounds that have induced random mutations and the polyploidization of chromosome sets.

Reply 2: Thank you for the comment and it was attended to accordingly. The figure legend was modified to avoid ambiguity or raise any questions regarding the prospects of ploidy on conferring legume resistance to salinity stress. According to literature, very few legumes, namely Arachis hypogaea, soybean and Medicago spp. (x2) as indicated in paragraph 2,3 (page 3) had allotetraploids/autotetraploids and the rest of polyploids are found in Poales or other non-leguminous taxa.

That is why such a Table could not be developed as this topic is very under researched and could contain spp. too few to represent the entire group of grain legumes as targeted in this paper.

To the best of our ability, and most importantly Table 1 could be developed showing widely available diploid legumes and their candidate genes that could be potentially duplicated for possible salinity stress tolerance.    

Comment 3: To add confidence to the authors arguments, it should be discussed in depth how genetic variations induced through random mutation by naturally or synthetically produced mutagenic compounds and the polyploidy of chromosome sets in leguminous crops can be maintained stably by the next generation.

Reply 3: Thank you for the comment and it was attended to accordingly. This comment was succinctly answered in paragraph 1 under section 2 on page 3.

Comment 4: Please revise the text to a more concise sentence.

Reply 4: Thank you for the comment and it was attended to accordingly. The entire manuscript was revised as suggested and we hold the view that the quality and standard of this manuscript matches both the Journal and researchers/students in the field.